# Designand Implementation of a Novel Wideband HF Communication System Based on NC-OFDM and Probabilistic Shaping

**DOI:** 10.3390/s25175596

**Published:** 2025-09-08

**Authors:** Rifei Yang, Yong Bai, Zhuhua Hu

**Affiliations:** School of Information and Communication Engineering, Hainan University, Haikou 570228, China; yrf@hainanu.edu.cn (R.Y.);

**Keywords:** NC-OFDM, probabilistic shaping, CCDM, LDPC, wideband HF

## Abstract

This paper proposes a novel wideband high-frequency (HF) communication system supporting video transmission based on non-contiguous orthogonal frequency division multiplexing (NC-OFDM) and probabilistic shaping (PS). The HF spectrum is currently very crowded; to find a free continuous frequency band around 500 KHz for video transmission is almost impossible. So this paper investigates how to exploit spectrum holes in the HF band with NC-OFDM technology. We propose a transmission scheme over a wideband HF channel modeled by the Institute for Telecommunication Sciences (ITS) channel model with valid bandwidth up to 1 MHz. In order to improve the reliability of proposed scheme, this paper further investigates the probabilistic shaping-based coding modulation. Simulation results show that the designed wideband HF NC-OFDM communication system can meet the data rate required for video transmission. In addition, the probabilistic shaping-based coding modulation provides a significant performance improvement over uncoded systems and the probabilistic shaping offers an extra 0.6 dB shaping gain in the wideband HF channel compared to equal probability constellation systems.

## 1. Introduction

HF communications transmit electromagnetic waves by reflection of the ionosphere, which can support long-distance communication without relays. As the ionosphere is basically indestructible, HF communications occupy an important position in military communications, and they have a wide range of applications in maritime communications and point-to-point inter-island communications [1]. The frequency range of HF communications, commonly covering between 3 to 30 MHz, is relatively narrow, so the HF communications have been limited to low speed and narrow band range for a long time, for example, the 3 KHz system bandwidth can only satisfy the needs of voice transmission. In recent years, the growing demand for high-speed HF communications in applications such as image and video transmission has driven the research of the next generation of HF communication systems, and one of the main research directions is wideband HF communication, which can be realized with higher data transmission rates [2].

The ionosphere is a transmission medium with layered, random variations. So the HF channels are frequency-selective and time-selective in nature. This makes the transmitted signals suffer from attenuation, and time and frequency dispersion [3]. The HF communication system based on orthogonal frequency division multiplexing (OFDM) is proposed in [4], which improves the system’s ability to resist frequency-selective fading and increases the spectrum efficiency, but the system bandwidth is only 3 KHz.

With the proposal of wideband HF channel models such as Volger and ITS [5,6], there has been considerable recent interest in wideband HF communications. A number of continuous-OFDM (C-OFDM) schemes are proposed in [7,8,9]. The system bandwidth has reached 24 KHz and above, and their performance is much better than the single carrier schemes [10]. However, these C-OFDM schemes still use static subcarrier allocation instead of dynamic subcarrier allocation.

The HF band is very crowded at present, so how to improve spectral efficiency to meet the requirements of high-speed data transmission has become a hot research topic. Researchers have found that the HF band exhibits “porosity” in both frequency and time; in other words, the HF band is divided into many available discontinuous sub-bands by interference from various users. Applying cognitive radio (CR) concepts to the HF communications is proposed in [11]. Researchers have developed a CR-based hidden Markov model to predict primary user activities and improve the performance of HFDVL systems in [12]. The method for monitoring the available HF band has been developed in [13].

A cognitive radio system based on NC-OFDM, which is a variant of OFDM that can effectively improve spectral efficiency and avoid channel interference, is presented in [14]. Then, NC-OFDM was applied to achieve discontinuous channel bonding in the RWAN communication systems of the IEEE 802.22 standard [15]. The NC-OFDM scheme has also been applied in satellite communications [16], but there are significant differences in channel characteristics, system bandwidth, and application scenarios between satellite communications and HF communications.

To our knowledge, the combination of NC-OFDM with HF communications has rarely been reported. In order to implement high-rate video transmission of the H.264 standard, we propose a novel wideband HF communication system based on NC-OFDM. It uses a suitable dynamic spectrum sensing scheme to search for the holes in the HF band and uses a subcarrier allocation algorithm to combine discontinuous subcarrier blocks, which are called as sub-bands, to provide a sufficient transmission bandwidth. For comparison, the spectral efficiency of the existing solutions are shown in Table 1.

How to construct reliable communication is one of the important challenges for HF communications. Both theoretical research and practice have proved that channel coding is an effective method to improve transmission reliability. In recent years, low-density parity-check (LDPC) codes have been widely studied for their excellent performance close to Shannon’s limit. Ref. [17] applied LDPC codes to HF communications and designed the MLC-LDPC-OFDM system, which improves the performance of the system by adjusting the code rate according to the characteristics of the HF channel. Ref. [18] further proposed the collaborative LDPC decoding algorithm for multi-base stations in an LDPC-OFDM system for HF communications.

High-order modulation (e.g., high-order QAM) improves the spectral efficiency of the system, but the uniformly distributed QAM constellation has shaping losses. To address this problem, a method called probabilistic shaping (PS) for fiber optic communication systems under Gaussian channel has been investigated in [19,20]. PS converts the uniformly distributed constellation points to an equally spaced, but unequally probability distributed constellation. The unequal distribution reduces the average power of the constellation symbols; in other words the equivalent minimum Euclidean distance of the constellation points increased under the condition of the same transmit power, and the noise immunity is enhanced. The probabilistic shaping coded modulation (PS-CM) system combines probabilistic-shaped high-order modulation with forward error correction (FEC) coding. It connects the distribution matcher and FEC encoder in series, and matches different rates and spectral efficiencies through the code rate of distribution matcher and FEC coder. Therefore, the spectral efficiency and reliability of the system have been improved. Currently, PS-CM systems have received a great deal of attention [21].

The main contributions of this paper are summarized as follows: In this paper, we propose an NC-OFDM scheme to ensure sufficient transmission bandwidth for wideband HF communications in the presence of various interference. To improve the reliability of the system, a combination of quasi-cyclic LDPC and PS is proposed. We provide a system design scheme for wideband HF video transmission between distant islands and use the ITS channel model, which supports bandwidth up to 1 MHz to simulate and verify the system design. The simulation results show that it can meet the requirements of the data transmission rate for video transmission and has a significant improvement in the system performance in terms of BER compared to the conventional HF communication systems without channel coding and PS.

The rest of this paper is organized as follows. In Section 2, we give the overview block diagram of the proposed scheme and present the compositions and the functions of each module; Section 3 depicts the principle of the proposed scheme such as dynamic spectrum sensing, NC-OFDM, QC-LDPC, PS, etc.; Section 4 presents the system simulation method and parameter setting and gives the simulation results and discussion; and Section 5 concludes this paper.

## 2. System Model

Figure 1 presents the HF communication scenario between the land and island; the transmitter and receiver are both fixed. Since it is a long-distance transmission range from 300 km to 2000 km, the line-of-sight and surface-wave propagation are not considered, only the sky-wave propagation is studied, and the channel is a transmission channel through the ionosphere.

Figure 2 shows the block diagram of the system, which is described in detail in the following.

### 2.1. Transmitter

System modules at the transmitter side are shown in the upper part of Figure 2. The input binary information bit sequence is divided into two parts; one is used for low-order modulation and encoded by an LDPC encoder directly, and the other is transferred to a PS-CM module, which is mainly composed of a distribution matcher and an LDPC encoder. The distribution matcher converts the uniformly distributed sequence of binary bits into symbols following certain probability distribution, then the symbols undergo binary mapping and channel coding, and then the non-uniformly distributed systematic bits and uniformly distributed check bits are modulated jointly by a high-order modulator.

The NC-OFDM adaptive transmission system joints a subcarrier bit and power allocator and a dynamic subcarrier allocator. The subcarrier bit and power allocation is performed using the feedback channel state information (CSI) from channel estimation to improve transmission efficiency. The subcarrier allocator knows the spectrum holes from the spectrum sensing and also uses the CSI. The output of the subcarrier allocator is converted into discrete time-domain sample signals through serial-to-parallel conversion, IFFT, parallel-to-serial conversion, and CP insertion. Finally, it enter the HF channel through the D/A converter.

### 2.2. Receiver

The modules of the receiver are shown in the lower part of Figure 2. The transmitted signal reaches the receiver through an HF channel, and is first converted into discrete time-domain sample signals by an A/D converter. Then the time–frequency resource grids are obtained through a series of processes such as removing cyclic prefix, serial-to-parallel conversion, FFT, and parallel-to-serial conversion, which are also the same as the ordinary OFDM. The next step is subcarrier extraction, which is the inverse process of the transmit side. The subcarrier extraction requires the position of active subcarriers from the transmitter; it extracts the data symbols from active subcarriers.

Let $X_{i}$ be the i-th transmitted complex symbol, then the received complex symbol after the above NC-OFDM transmitting, HF channel and NC-OFDM receiving can be expressed as(1)$$R_{i}=h_{i}\cdot X_{i}+n_{i}$$
where $h_{i}$ is the complex channel frequency response on the corresponding subcarrier and $n_{i}$ denotes complex Gaussian white noise with mean 0 and variance $2\sigma^{2}$. If the ideal channel estimation is used, the received symbol after channel equalization can be expressed as(2)$$Y_{i}=\frac{R_{i}}{h_{i}}=X_{i}+\frac{n_{i}}{h_{i}}$$

Let $n_{i}^{\prime }=n_{i}/h_{i}$, then $n_{i}^{\prime }$ is still a complex Gaussian variable with mean 0 and variance $2\sigma^{2}/{\left|h_{i}\right|}^{2}$ [22].

The inverse PS-CM module performs the inverse process at the receiver side, which consists internally of the bitwise soft demodulator, the channel decoder, the demapper, and the dematcher in order. This module outputs the data of high-order modulation, and then combines it with the data of low-order modulation for output.

## 3. Principle of the Proposed Scheme

### 3.1. Dynamic Spectrum Sensing

Current research has revealed that there are many spectrum holes in the 16 MHz to 23 MHz range of the HF channels near the equator, which can meet the bandwidth requirements of the system. Therefore, we have chosen this frequency band as the sensing frequency band. There are many algorithms for spectrum sensing [23], such as energy detection, wavelet detection, matched filters, etc. HF channels are quickly variable, so the spectrum sensing algorithm needs to be in real-time. There are many possible interferences in HF channels, and the prior information of the interference signals cannot be obtained normally.

We propose to integrate the simple two-stage spectrum sensing scheme based on energy detection [24] in HF communications. The block diagram of the scheme is given in Figure 3. In the first stage, we divide the sensing frequency band from 16 MHz to 23 MHz into 7 subchannels with 1 MHz intervals. The received signal is sampled by A/D and then filtered by the bandpass filters individually. The average power of the filtered signal is then calculated for these subchannels. Then the subchannels with lower average power are selected as alternative. The subchannels with strong interference will be excluded by a higher threshold. We refer to this stage as the coarse scanning. In the second stage, we select the subchannel with the lowest average power, and down-shift the signal in this subchannel to 0 to 1 MHz baseband, then detect it using a periodogram energy detection algorithm to obtain the power spectral density of the signal. Finally, all holes in the subchannel are found by comparing with the adaptive threshold. This second stage is referred to as fine scanning. If the total bandwidth of the holes in the subchannel is not sufficient for data transmission, we will switch to another alternative subchannel to perform the second stage of detection, and so on.

In general, the spectrum detection problems of the received signal can be interpreted as a binary hypothesis:(3)$$\left\{\begin{array}{c} H_{0}:\;r\left(t\right)=w\left(t\right)\\ H_{1}:\;r\left(t\right)=h\left(t,\tau \right)⊗s\left(t\right)+w\left(t\right) \end{array}\right.$$
where $H_{0}$ means that there is no interference signal, while the hypothesis denoted as $H_{1}$ means the interference signal $s\left(t\right)$ is present. The $h\left(t,\tau \right)$ is the channel impulse response at time t and $w\left(t\right)$ is AWGN.

The signal after A/D sampling is $r\left(n\right)$. The average power of subchannel can be given as follows:(4)$$P_{sub}=\frac{1}{N_{sp}}\sum_{n=0}^{N_{sp}-1}{\left|r_{bp}\left(n\right)\right|}^{2}$$
where $r_{bp}\left(n\right)$ denotes the filtered signal from the bandpass filter, and $N_{sp}$ is the number of sampling points. The subchannels with $P_{sub}>\zeta$ will be excluded from the alternatives, where ζ is the threshold value.

The estimated power spectral density (PSD) can be defined as(5)$$\hat{S}\left(\omega \right)=\frac{1}{M}{\left|\tilde{R}\left(\omega \right)\right|}^{2}=\frac{1}{M}{\left|\sum_{m=0}^{M-1}{\tilde{r}}_{bp}\left(m\right)e^{-j\omega m}\right|}^{2}$$
where ${\tilde{r}}_{bp}\left(m\right)$ is the baseband filtered signal in the subchannel and M is the Discrete-Time Fourier Transform (DTFT) size. Welch’s method is adopted for reducing the frequency leakage and improving the accuracy, and the variance in PSD estimation can be further reduced by averaging multiple data segments:(6)$$S\left(\omega \right)=\frac{1}{K}\sum_{i=1}^{K}{\hat{S}}_{i}\left(\omega \right)$$
where K is the number of data segments.

From the above analysis, the two-stage spectrum sensing scheme we proposed does not need the prior information and is based on Fourier transform that can be quickly implemented by FFT. Furthermore, by dividing the sensing frequency band into multiple subchannels, the scheme reduces the number of FFT points while maintaining the detection accuracy. Therefore, it has low complexity, is fast, and performs well in real-time.

### 3.2. NC-OFDM

NC-OFDM is an extension of OFDM, which can select discontinuous subcarriers for data transmission. This dynamic use of the spectrum is more flexible compared to OFDM, and it more easily meets the system’s bandwidth requirements.

The OFDM system divides the channel into a number of subcarriers for data transmission. In the traditional OFDM system, the data is continuously distributed in the time–frequency resource grid, but in the NC-OFDM system the data is not continuous in the frequency domain. The transmitter of the NC-OFDM system adds a dynamic subcarrier allocator. It knows which subcarriers are available or the spectrum holes from spectrum sensing, and then does not send data on the unavailable subcarrier, which is equivalent to switching off the interfered subcarriers. The subcarrier deallocator in the receiver knows which subcarriers are active from the transmitted subcarrier allocation information.

The OFDM time domain equivalent baseband complex signal can be expressed as(7)$$x\left(t\right)=\sum_{k=0}^{Nc-1}a_{k}^{\left(m\right)}\exp\left(j2\pi k\Delta ft\right)$$
where $N_{c}$ is the number of subcarriers, i.e., the number of IFFT/FFT points, $a_{k}^{\left(m\right)}$ denotes the data symbol transmitted on the k-th subcarrier of the m-th OFDM symbol, m=1,2,⋯,M, *M* is the number of OFDM symbols per frame, and kΔf denotes the subcarrier frequency.

The implementation principle of NC-OFDM is as follows: let there be N data symbols in each frame after modulation, where the i-th data symbol is denoted by $X_{i}$, i=1,2,⋯,N, $X_{i}$ is complex number in the low-pass equivalent system. Compared to the traditional OFDM, the subcarrier allocator divides the $N_{c}$ subcarriers into two subsets: I={interferedsubcarriers} and UI={availablesubcarriers}. Data symbols are assigned to subcarriers as follows:(8)$$a_{k}^{\left(m\right)}=\left\{\begin{array}{c} 0,\left\{k\left|k\in I\right.\right\}\\ X_{i},\left\{k\left|k\in UI\right.\right\} \end{array}\right.$$

The 0 here means no data and does not refer to binary bit 0. If the number of interfered subcarriers is $N_{I}$ and the number of available subcarriers is $N_{c}-N_{I}$, then it has to satisfy $(N_{c}-N_{I})*M>N$ in order to send out N data symbols in one frame. Since the number of available subcarriers is less than the total number of subcarriers, transmitting the same number of data symbols requires more subcarriers than ordinary OFDM, which means that the size of the IFFT/FFT is increased. This inevitably increases the computation complexity of IFFT/FFT. However, since NC-OFDM has no data on interfered subcarriers, this makes the IFFT/FFT transform of NC-OFDM sparse. Some low-complexity IFFT/FFT transform algorithms when data is sparse have been researched, which add only marginal complexity [14]. In summary, the NC-OFDM system is able to increase a small amount of system complexity to solve the problem of how to obtain a larger bandwidth when the HF band has interference.

We propose a lightweight adaptive joint subcarrier allocation and bit power allocation scheme for highly dynamic ionospheric HF channel. This scheme is applicable to single-user mode NC-OFDM systems.

Spectrum sensing finds the locations of all holes in the 1 MHz frequency band. The dynamic subcarrier allocation algorithm should fully utilize the subcarriers with better channel condition within holes to improve the transmission efficiency. However, if the allocation strategy is to select the subcarriers with the best channel condition, it will lead to the dispersion of allocated sub-bands. Since the guard band is inserted at both ends of the sub-band to prevent interference, the dispersion of the sub-band will decrease spectral efficiency. Therefore, we adopt the suboptimal allocation algorithm using one hole as the allocation unit, which can reduce computational complexity and improve real-time performance. The algorithm proceeds as follows.

Assume that the current CSI is available; the Shannon capacity of the k-th subcarrier is calculated according to(9)$$C\left(k\right)=\Delta f\log_{2}\left(1+{\left|H_{k}\right|}^{2}\frac{P_{k}}{N_{0}}\right)$$
where Δf, $H_{k}$, and $P_{k}$ represent subcarrier spacing, channel frequency response, and transmission power at the kth subcarrier, respectively.

The average channel capacity of one hole is(10)$$\bar{C}=\frac{1}{K_{h}}\sum_{k=0}^{K_{h}-1}C\left(k\right)$$
where $K_{h}$ is the number of subcarriers in one hole.

Firstly, we use Equation (Equation 10) to calculate the average channel capacity of each hole. Then all holes are sorted in descending order by average channel capacity. Then the holes are sequentially allocated to the sub-bands in order until the sum of all subcarrier bandwidths in the allocated sub-bands satisfies the data transmission bandwidth requirement. This algorithm does not have an iterative process; the overall complexity is dominated by multiplications, additions, and comparisons. The computation complexity of subcarrier channel capacity is $O\left(K_{ht}\right)$, where $K_{ht}$ represents the total number of subcarriers in all holes. The complexity of average channel capacity is $O\left(K_{h}\right)$, and the complexity of sorting is $O\left(N_{h}\right)$, where $N_{h}$ is the number of holes.

Subcarrier bit and power allocation adaptively optimize resource allocation by using CSI in the transmitter. In practical HF systems, the transmit power is strictly limited. Therefore, the margin adaptive (MA) criterion can be adopted. The MA criterion aims to minimize total transmit power under constraints of a target data rate and BER. For the large number of subcarriers in our wideband HF NC-OFDM system, if bit and power allocation are calculated for each subcarrier, the computational complexity is too high to meet the real-time requirements. To address this challenge, we propose to use the existing simple blockwise loading algorithm (SBLA) algorithm [25], which groups adjacent subcarriers into blocks, and the subcarriers in one block adopt the same modulation mode. For the principles and implementation details of this algorithm, refer to [25]. After the bit and power allocation of the SBLA algorithm, the subcarriers with high SNR allocate more bits than the subcarriers with low SNR.

Building upon existing SBLA algorithms, we propose the subcarriers modulated with low-order modulation scheme, such as BPSK, QPSK, 8PSK, etc., to be encoded by the LDPC encoder directly. Conversely, the subcarriers modulated with high-order modulation, such as 16QAM, 32QAM, 64QAM, etc., to be processed by the PS encoder. This can further reduce the BER of the high-order modulation by achieving shaping gain, thus improving the overall system performance.

Because there is no iterative process, the SBLA algorithm reduces the computational complexity significantly. Moreover, the larger the block size, the lower the computational complexity. However, the performance will also correspondingly drop.

The high Peak-to-Average Power Ratio (PAPR) is a major problem in OFDM systems as well as in NC-OFDM systems [15]. In NC-OFDM systems, the complementary cumulative distribution function (CCDF) of PAPR can be evaluated by(11)$$\mathrm{Pr}\left\{PAPR>z\right\}=1-{\left(1-e^{-\frac{z}{\sigma^{2}}}\right)}^{N}$$
where N is the number of active subcarriers. $\sigma^{2}$ is the variance of OFDM data symbols. This formula indicates that the PAPR is related to the number of active subcarriers rather than the number of FFT points in NC-OFDM systems. The following reasons may affect the PAPR performance: Firstly, due to the fact that NC-OFDM systems have many null subcarriers, the input symbols are not identically and independently distributed (i.i.d.). Secondly, the position of active subcarriers will affect the PAPR. A number of techniques have been proposed to deal with the PAPR problem. Among them, selective mapping (SLM) is a kind of probabilistic technique that can be applied to NC-OFDM systems. We will compare the PAPR performance of NC-OFDM and OFDM systems later in the simulation section.

### 3.3. Probabilistic Shaping Coded Modulation

The distribution matcher transforms Bernoulli distributed input bits into output symbols that follow the expected distribution. In this paper, we use constant composition distribution matching (CCDM) proposed in [26]. The CCDM is reversible and has fixed input and output length, whose functional block diagram is shown in Figure 4.

Let $u^{k}=u_{1}\cdots u_{k}$ be the input binary bit sequence of length k, following a Bernoulli distribution of order k. The code rate of the distribution matcher, R=k/n, denotes the number of bits carried per symbol. After mapping by the distribution matcher, the output is a symbol sequence $A^{n}=A_{1}\cdots A_{n}$, of length n, where $A_{j}\in \left\{1,\cdots ,M\right\}$, j=1,2⋯n. The symbols are mutually independent and follow Maxwell–Boltzmann distribution; the probability is $P_{A}$. So the output sequence emulates a memoryless discrete source $P_{A}$, and the two are very close in information dispersion. The output sequences of CCDM are n-class sequences, i.e., among all the output sequences of length n, the output sequences satisfy(12)$$P_{A(a)}=\frac{n_{a}}{n},a\in \left\{1,\cdots ,M\right\}$$(13)$$\sum_{a=1}^{M}n_{a}=n$$
where $n_{a}$ denotes the number of occurrences of symbol a in the sequence, and $P_{A\left(a\right)}$ denotes the probability that symbol A takes a. Obviously in any sequence of class n, the number of occurrences of each symbol is constant and hence it is called a constant composition distribution matcher. $T_{P_{A}}^{n}$ denotes the set of n-class sequences, according to the theory of permutations and combinations; it can be obtained as follows:(14)$$\begin{array}{c} \left|T_{P_{A}}^{n}\right|=\left(\begin{array}{c} n\\ n_{1} \end{array}\right)\left(\begin{array}{c} n-n_{1}\\ n_{2} \end{array}\right)\cdots \left(\begin{array}{c} n-n_{1}\cdots -n_{k-2}\\ n_{k-1} \end{array}\right)\left(\begin{array}{c} n-n_{1}\cdots -n_{k-1}\\ n_{k} \end{array}\right)\\ =\frac{n!}{n_{1}!n_{2}!\cdots n_{k-1}!n_{k}!} \end{array}$$

Because the distribution matcher requires reversibility, the input binary sequence length k should be no greater than $\log_{2}^{\left|T_{P_{A}}^{n}\right|}$. We take the input length to be $m=\left\lfloor \log_{2}\left|T_{P_{A}}^{n}\right|\right\rfloor$, then the coding function of CCDM can be defined as(15)$$f_{ccdm}:{\left\{0,1\right\}}^{m}\to T_{P_{A}}^{n}$$

The codebook for the CCDM code is $C_{ccdm}=f_{ccdm}\left({\left\{0,1\right\}}^{m}\right)$, the codebook size is $\left|C_{ccdm}\right|=2^{m}$.

The encoding and decoding of CCDM can be achieved by the improved arithmetic coding algorithm introduced in [27,28]. The algorithm calls n-classes of sequences as source sequences, denoted as $A=\left\{a_{1},a_{1},\cdots ,a_{i},\cdots ,a_{n}\right\}$, and the finite-length prefix sequence of A is denoted by α, e.g., the prefix of the symbol $a_{i}$ is $\alpha =\left\{a_{1},a_{2},\cdots ,a_{i-1}\right\}$, and the statistical properties of the source sequences can be represented by the conditional probability $P\left(a_{i}|\alpha \right)$, which can be obtained from the distribution $P_{A\left(a\right)}$. Initially, the unit interval [0, 1] is corresponded to the empty sequence λ, and with the increase in source sequence symbols, the interval is subdivided into sub-intervals continuously, and the width of the sub-intervals is corresponded to the probability of the sequence λ. After n times of subdividing, the intervals of the source sequences A are equal, and the width of sub-interval corresponding to A is $1/\left|T_{P_{A}}^{n}\right|$. The idea of the encoding algorithm is to consider the input binary sequence as an expansion of binary decimal numbers. The point represented by this binary decimal number is located between [0, 1), and then we can recursively determine which sub-interval sequence the point lies in, until it is determined that the point lies in the sub-interval corresponding to the source sequence A, then the encoded symbol sequence A is obtained. The decoding algorithm also uses recursive methods, starting from the symbol sequence, i.e., the source sequence A, and passing through a recursive of subdividing the interval until the starting position and width of the sub-interval corresponding to A is obtained, a point in the sub-interval is selected, and the positional value of this point is expressed in the form of a binary decimal expansion, thus obtaining the binary input sequence.

In this paper, one-dimensional constellation shaping with high-order modulation is used, i.e., two real amplitude shift keying (ASK) symbols are mapped into one complex QAM symbol. At this point when $2^{m}$-QAM modulation is used (m denotes the number of bits corresponding to each QAM constellation symbol), the first m/2 bits of the QAM symbol bit sequence $B_{0}B_{1}\cdots B_{m-1}$ correspond to real numbers on the I-axis of the I/Q plane, and the last m/2 bits correspond to real numbers on the Q-axis. Here, $B_{0}$ and $B_{m/2}$ are sign bits and the rest are amplitude bits, the number of amplitude bits is m−2, and there are $2^{\left(m/2-1\right)}$ types of amplitude.

The PS-CM uses a probability amplitude shaping (PAS) scheme, which is a series connection of the distribution matcher (DM) and the FEC encoder, so that the amplitude of the constellation points after QAM modulation follows the Maxwell–Boltzmann distribution. The block diagram of the PAS scheme is shown in Figure 5.

The information source generates a uniformly distributed information bit sequence $u^{k}=u_{1}\cdots u_{k}$, which passes through a distribution matcher with a code rate of $R_{CCDM}=k/n$. The output is an amplitude sequence $A^{n}=A_{1}\cdots A_{n}$, with unequal distribution and it emulates a discrete memoryless source $P_{A}$, where $A_{j}\in \left\{1,\cdots ,M\right\},j=1,2,\cdots ,n,M=2^{\left(m/2-1\right)}$. The function $\beta \left(•\right)$ maps the amplitude sequence into the amplitude bit sequence $a_{b}$, and the length is n*(m/2−1). $a_{b}$ is then encoded by the LDPC encoder with code rate $R_{LDPC}=\left(m/2-1\right)/\left(m/2\right)$, and the systematic bits of the LDPC output codeword are still the amplitude bit sequence $a_{b}$, and the length of the check bit sequence $s_{b}$ is n, which is used as the sign bit sequence. According to the LDPC coding theory, check bits are approximately equally distributed, so the sign bit sequence is approximately uniformly distributed. Sequence $a_{b}$ and $s_{b}$ are recombined by the bit combination module, one bit in $s_{b}$ and m/2−1 bits in $a_{b}$ are combined as sequence $B^{m/2}=B_{0}B_{1}\cdots B_{\left(m/2\right)-1}$, and the output $QAM_{bin}$ contains n sub-sequences of $B^{m/2}$. The $QAM_{bin}$ is then modulated by the QAM modulator to obtain the constellation symbol sequence $X^{n/2}$; $X^{n/2}$ is a complex sequence of length n/2.

### 3.4. Soft Demodulation Algorithm

Since we use one-dimensional constellation shaping for our QAM modulation, the i-th received symbol $Y_{i}=Y_{R_{i}}+jY_{I_{i}}$, where the real part $Y_{R_{i}}$ corresponds to the received in-phase bits $B^{I}=B_{0}B_{1}\cdots B_{j}^{I}\cdots B_{\left(m/2-1\right)}$ with a length of m/2, and the imaginary part $Y_{I_{i}}$ corresponds to the quadrate bits $B^{Q}=B_{\left(m/2\right)}B_{\left(m/2+1\right)}\cdots B_{j}^{Q}\cdots B_{m-1}$ and also has a length of m/2.

The bitwise soft demodulation algorithm is to calculate the log-likelihood ratio (LLR) soft bit values by received $Y_{R_{i}}$ and $Y_{I_{i}}$. According to the definition of LLR, the formulas for calculating LLR values of the bit $B_{j}^{I}$ and the bit $B_{j}^{Q}$, respectively, are(16)$$LLR\left(B_{j}^{I}\right)=\ln\left[\frac{P_{r}\left(B_{j}^{I}=0\left|Y_{Ri}\right.\right)}{P_{r}\left(B_{j}^{I}=1\left|Y_{Ri}\right.\right)}\right]=\ln\left[\frac{\sum_{B^{I}\in {\left\{0,1\right\}}^{\frac{m}{2}};B_{j}^{I}=0}\exp\left(-{\frac{{\left|h_{i}\right|}^{2}\left(Y_{Ri}-x_{B^{I}}\right)}{2\sigma^{2}}}^{2}\right)P_{r}\left(x_{B^{I}}\right)}{\sum_{B^{I}\in {\left\{0,1\right\}}^{\frac{m}{2}};B_{j}^{I}=1}\exp\left(-{\frac{{\left|h_{i}\right|}^{2}\left(Y_{Ri}-x_{B^{I}}\right)}{2\sigma^{2}}}^{2}\right)P_{r}\left(x_{B^{I}}\right)}\right]$$(17)$$LLR\left(B_{j}^{Q}\right)=\ln\left[\frac{P_{r}\left(B_{j}^{Q}=0\left|Y_{Ii}\right.\right)}{P_{r}\left(B_{j}^{Q}=1\left|Y_{Ii}\right.\right)}\right]=\ln\left[\frac{\sum_{B^{Q}\in {\left\{0,1\right\}}^{\frac{m}{2}};B_{j}^{Q}=0}\exp\left(-{\frac{{\left|h_{i}\right|}^{2}\left(Y_{Ii}-x_{B^{Q}}\right)}{2\sigma^{2}}}^{2}\right)P_{r}\left(x_{B^{Q}}\right)}{\sum_{B^{Q}\in {\left\{0,1\right\}}^{\frac{m}{2}};B_{j}^{Q}=1}\exp\left(-{\frac{{\left|h_{i}\right|}^{2}\left(Y_{Ii}-x_{B^{Q}}\right)}{2\sigma^{2}}}^{2}\right)P_{r}\left(x_{B^{Q}}\right)}\right]$$
where $P_{r}\left(x_{B^{I}}\right)$ is the probability of transmission the real number $x_{B^{I}}$ corresponding to the $B^{I}$. For general soft demodulation, the $P_{r}\left(x_{B^{I}}\right)$ term is eliminated up and down since $x_{B^{I}}$ is transmitted with equal probability, but for the PS system, due to the unequal probability of $x_{B^{I}}$ transmitted, the equations need to be multiplied by this factor. We call the above equation the exact LLR formulation under PS.

The soft bits of LLR are passed to the LDPC decoder, which employs a belief propagation algorithm (BP). This algorithm performs decoding by passing and iterating belief messages.

After decoding by LDPC, the systematic bits act as the amplitude bits, which are then mapped into the amplitude sequence ${\hat{A}}^{n}$ by the inverse mapping function $\beta^{-1}\left(•\right)$, and then decoded into the received information bits by the inverse distribution matching $DM^{-1}$, thus completing the whole receiving process.

## 4. Simulation Results and Discussions

The communication scenario studied in this paper is the long-distance communication between islands, and the communication distance is from 300 to 2000 km. Since it is a long-distance transmission, we only consider the sky-wave propagation, and the HF signals arrive at the receiver through the reflection of the ionosphere by one hop. The HF band from 2 MHz to 16 MHz is occupied by civil broadcasting, so the HF band we can use is between 16 MHz and 23 MHz. The maximum bandwidth of the system is 1 MHz; by means of NC-OFDM technology, the actual data bandwidth can reach 500 KHz.

The channel model adopts the wideband HF ITS channel model [5,6], which is based on long-term actual test data and describes the statistical properties of channel delay power profile, Doppler shift, and Doppler spread through three modules, which can reflect the transmission characteristics and structural properties of the actual channel in a more comprehensive way. We use the channel parameter settings of 126 km long path in mid-latitude given by ITU-9C/37-E in our simulation.

According to the maximum delay extension of the channel, the coherence bandwidth of the channel is Bc=11τdτd=1170μs≈14KHz70μs≈14KHz. We choose the subcarrier interval of 39.0625 Hz, which is much smaller than the coherence bandwidth of the channel, and it can effectively counteract the channel’s frequency selectivity, i.e., the channel basically maintains a flat fading in one subcarrier interval of the frequency domain, so that the receiving end can be able to receive by linear equalization. Also, because the total bandwidth of the system is 1 MHz ≫ channel coherence bandwidth of 14 KHz, the frequency equalization is required between the subcarriers. The OFDM symbol period of $T_{sym}$ = 25.6 ms is much smaller than the coherence time of the channel of 10 s, so the signal will not undergo time-selective fading in one OFDM symbol period. The time-selective fading between OFDM symbols within a frame can be resolved by inserting pilot signal and channel estimation. The length of the cyclic prefix (CP) is 1/4 OFDM symbol period, i.e., 6.4 ms, which is greater than the channel multipath maximum delay. We assume perfect channel information in simulating.

The NC-OFDM scheme is simulated using the following method. Firstly, the number of FFT points is set to 25,600, and the total number of active subcarriers is 12,800. Next, we assume that the interference is random occurrence in the HF channel. We detect the holes by means of spectrum sensing. After that, we divide the 12,800 active subcarriers into several sub-bands and assign to the holes dynamically.

We use quasi-cyclic LDPC codes with code length 64,800, which is compliant with DVB-S.2 standard; LDPC code rate $R_{LDPC}$ = 4/5, LDPC decoding is performed by the belief propagation algorithm, and the number of iterations is set to 10.

We use an improved arithmetic coding algorithm for the PS encoding and decoding, and use scaling and rounding techniques for encoding, which essentially implements the floating point representation through integer processing. The complexity of the improved arithmetic coding algorithm is reduced from $O\left(n^{2}\right)$ to $O\left(n\right)$. The code rate of the distribution matcher is set according to the modulation order, and the relationship between the total system code rate and the LDPC code rate and the distribution matcher code rate is given in the following equation:(18)$$R=\frac{R_{CCDM}+1}{m}+R_{LDPC}-1$$
where R is the total system code rate, $R_{CCDM}$ denotes the CCDM distribution matcher code rate, $R_{LDPC}$ is the LDPC code rate, and m is half the number of binary bits corresponding to each constellation point of the QAM.

Using the Monte Carlo method, we employed Matlab (R2022a) tools to simulate the proposed NC-OFDM-and-PS-based HF communication system where the main simulation parameters are summarized in Table 2.

### 4.1. Dynamic Spectrum Sensing and NC-OFDM Signal Spectrum

We simulate the time-varying interference scenarios. Some BPSK signals with different carriers, bandwidths, and SNRs are generated to simulate the actual interference signals, and the noise is the AWGN. After the detection by the energy detection algorithm introduced in Section 3.1, the power spectral density of interference signals and the location of holes can be obtained, as shown in Figure 6. Then, the active subcarriers are mapped to these holes by the subcarrier assignment algorithm introduced in Section 3.2. The robustness of the system was tested by simulation. Figure 6 shows that when the burst interference comes, the spectrum sensing can detect the change of the channel in time, then the system performs subcarrier switching to avoid interference.

### 4.2. BER Performance Comparison of NC-OFDM and C-OFDM

In Figure 7, the total subcarriers channel capacity is presented for the dynamic subcarrier allocation scheme of the NC-OFDM system. As a benchmark, we compare with the fixed subcarrier allocation of C-OFDM. It can be seen that the channel capacity has been significantly increased under different SNR conditions. The average increase is 0.5 Mb/s/Hz or the SNR gain is 2 dB for the same channel capacity. The reason for the capacity increase is that, in the frequency-selective channel, the subcarriers with higher channel gains are prioritized dynamically.

The BER performance of the NC-OFDM system with joint adaptive subcarrier allocation and bit power allocation scheme is evaluated by simulations. The system parameters are listed in Table 2. The simulation results are shown in Figure 8. The results show that the BER performance of the C-OFDM system with greedy algorithm is superior to the C-OFDM system without adaptive resource allocation, but the NC-OFDM system with joint adaptive scheme obtains a better performance. Actually, there is about 1 dB SNR gain for NC-OFDM compared to C-OFDM when BER is equal to $10^{-2}$. The results indicate that the NC-OFDM scheme has greater potential than C-OFDM in terms of adaptive resource allocation.

### 4.3. PAPR Performance Comparison of NC-OFDM and C-OFDM

Figure 9 shows the PAPR performance of NC-OFDM and C-OFDM. The PAPR performance is evaluated in terms of CCDF. The performance comparison is carried out with a fixed number of active subcarriers and varying FFT sizes. It can be seen that the PAPR performance of NC-OFDM is slightly worse than that of C-OFDM; the reason is that the dynamic assignment of subcarriers increases the probability that multiple subcarrier signals add up coherently. The SLM algorithm can also be used in the NC-OFDM system to reduce PAPR; the result shows when $CCDF=10^{-2}$, the PAPR is reduced to 10.1 dB.

### 4.4. Performance Analysis of NC-OFDM with LDPC Coding in the HF ITS Channel

In order to study the performance of the LDPC-coded NC-OFDM system over an HF channel, we compare the BER performance without channel coding and with LDPC coding under different QAM modulation orders, and the simulation results are shown in Figure 10. It can be seen that the BER performance of the NC-OFDM system that uses quasi-cyclic LDPC coding is significantly improved compared to the NC-OFDM system without channel coding. For example, under conditions of 16QAM, code length 64,800, and 10 iterations, the LDPC codes with code rate 1/2 have a coding gain of about 9 dB at $BER=10^{-2}$ compared to the uncoded system, and the LDPC codes with code rate 1/2 have even a performance improvement of about 10.5 dB at $BER=10^{-2}$ in 64QAM. The simulation results also show the impact of different LDPC code rates. We simulated the LDPC code rates such as 1/2, 2/3, 3/4, etc. The BER performance improves as the code rate decreases, e.g., for 64QAM with $BER=10^{-4}$, the performance of LDPC codes with 1/2 code rate improves by 2 dB over LDPC codes with 2/3 code rate.

### 4.5. Performance Simulation of NC-OFDM+LDPC+PS in HF Channel

Figure 11a and Figure 11b show the effect of PS at 16QAM and 64QAM modulation, respectively. The height of the bar corresponds to the normalized probability of the constellation symbols appearing, from which it can be seen that the probability of the constellation symbols appearing with low symbol energies is much higher than symbols with high symbol energies, and the probabilistic distribution of the constellation points obeys the Maxwell–Boltzmann distribution [29]. The inward aggregation of the constellation points reduces the average power of the constellation points, which can improve the system’s immunity to interference in power-limited systems.

We first study the impact of probabilistic shaping on the BER performance in the AWGN channel. Figure 12a and Figure 12b show the BER curves for 16QAM with a total system code rate of 2/3 and 3/4, respectively. We compare the BER performance of LDPC coding only and probabilistically shaped LDPC coding under the same spectral efficiency condition. For example, when the system is LDPC and codes only without PS, the total system code rate equals the LDPC code rate and is 2/3. By comparison, the total code rate of the system with PS and LDPC is also 2/3, but the distribution matcher code rate is 11/15, and the code rate of LDPC is 4/5. From Figure 12, we can see that the BER performance of the system with the PS has been improved and the waterfall phenomenon is evident. The shaping gain at $BER=10^{-6}$ is about 0.6 dB in Figure 12a. Similarly, in Figure 12b, a shaping gain is about 0.5 dB at $BER=10^{-6}$.

Figure 13a and Figure 13b show the BER curves for 64QAM modulation in the AWGN channel with a total system code rate of 2/3 and 3/4, respectively. The results are similar to the case at 16QAM, where the use of probabilistic shaping gives an improvement in BER performance. As shown in Figure 13a, there is about 0.9 dB of shaping gain at $BER=10^{-6}$ for total system code rate 2/3. As shown in Figure 13b, at total system code rate 3/4, there is also about 0.9 dB of shaping gain at $BER=10^{-6}$. Comparing the performance gain results of 16QAM and 64QAM, it can be seen that probabilistic shaping can achieve higher gain under high-order modulation, and the higher the modulation order, the greater the gain.

Finally, we study the performance of probabilistic shaping in HF ITS channels. Figure 14 and Figure 15 show the simulation results under 16QAM and 64QAM, respectively. Due to HF channel fading, the noise and distortion experienced by the transmitting bits at the correlation time or in the correlation bandwidth are strongly correlated, thus requiring complex interleaving schemes. We assume a perfect interleaving in our simulations, which weakens the correlation of the bits entering the LDPC decoder and improves the BER performance. As can be seen from Figure 14, the performance of PS is worse than LDPC only when the SNR is low. This is because the LDPC code rate used in the PS is higher than that of LDPC only when comparing two systems with the same total system code rate, which is an important factor affecting the performance, and the higher the code rate, the worse the performance. The PS has better performance when the SNR is higher, as shown in Figure 14a. For 16QAM, 2/3 total system code rate has a shaping gain of about 0.6 dB at $BER=10^{-6}$. As shown in Figure 14b, for 16QAM, 3/4 total code rate has the same shaping gain of about 0.6 dB at $BER=10^{-6}$. Figure 15a and Figure 15b show the performance simulation results for 64QAM with total code rates of 2/3 and 3/4, respectively. At $BER=10^{-6}$, the shaping gain is about 0.8 dB, which reflects the characteristic of shaping gain increasing following the increase in modulation order.

## 5. Conclusions

In this paper, a wideband HF communication system based on NC-OFDM and PS is studied. The system is proposed to overcome the HF band interference problem of HF communications. NC-OFDM acquires the bandwidth to satisfy video transmission demands by binding discontinuous frequency bands. The method to improve the system transmission reliability is further studied and PS-CM is used in the system. We adopt the wideband HF ITS channel model to more accurately simulate and verify the performance of the system. The ITS model can support a channel bandwidth of 1 MHz. After theoretical analysis and simulation verification, the probabilistic shaping-based LDPC coding has improvement in BER performance compared to the equal probability LDPC coding in an HF channel. The research in this paper has certain reference value and significance for the design of wideband HF communication systems. 

## Figures and Tables

**Figure 1 sensors-25-05596-f001:**
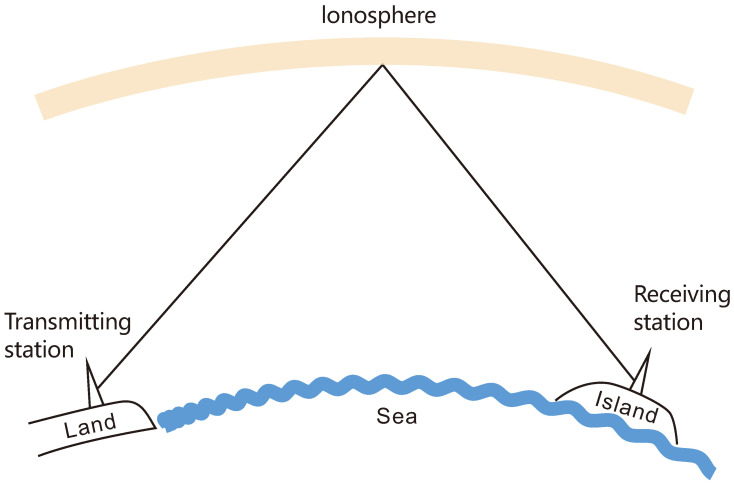
Communication scenario.

**Figure 2 sensors-25-05596-f002:**
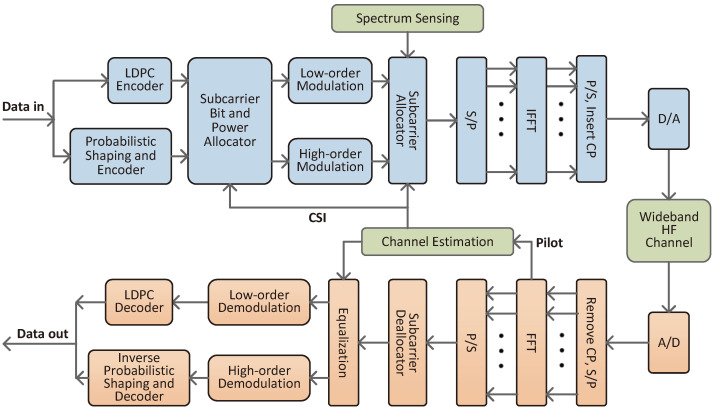
System block diagram.

**Figure 3 sensors-25-05596-f003:**
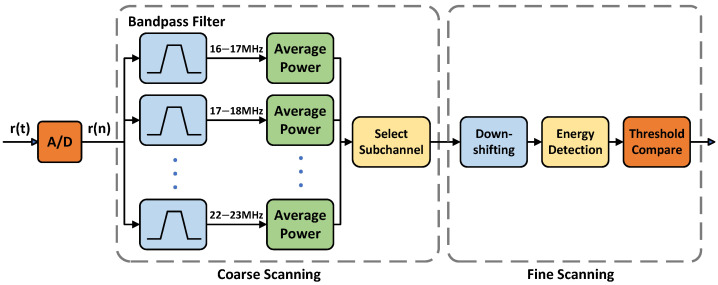
Block diagram of the two-stage spectrum sensing scheme.

**Figure 4 sensors-25-05596-f004:**
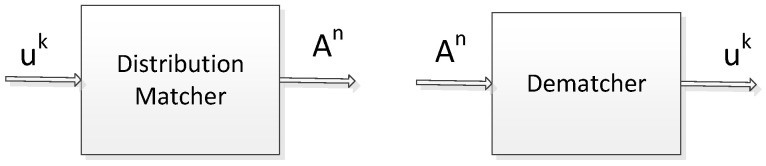
Block diagram of distribution matcher and dematcher.

**Figure 5 sensors-25-05596-f005:**
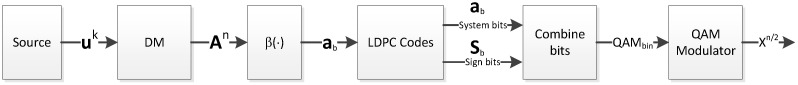
Block diagram of the PAS scheme.

**Figure 6 sensors-25-05596-f006:**
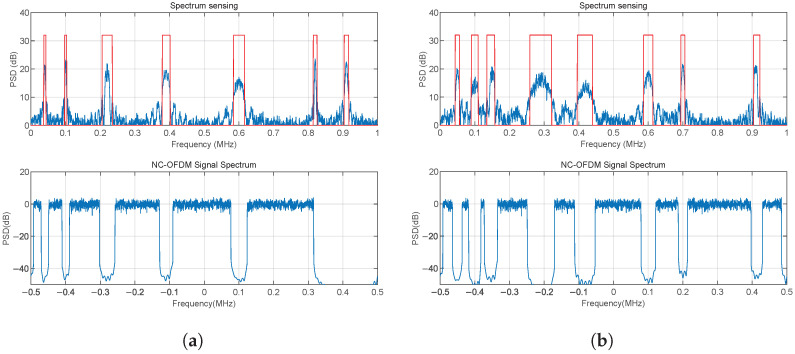
Spectrum sensing results and subcarrier switching: (**a**) Scenario 1. (**b**) Scenario 2.

**Figure 7 sensors-25-05596-f007:**
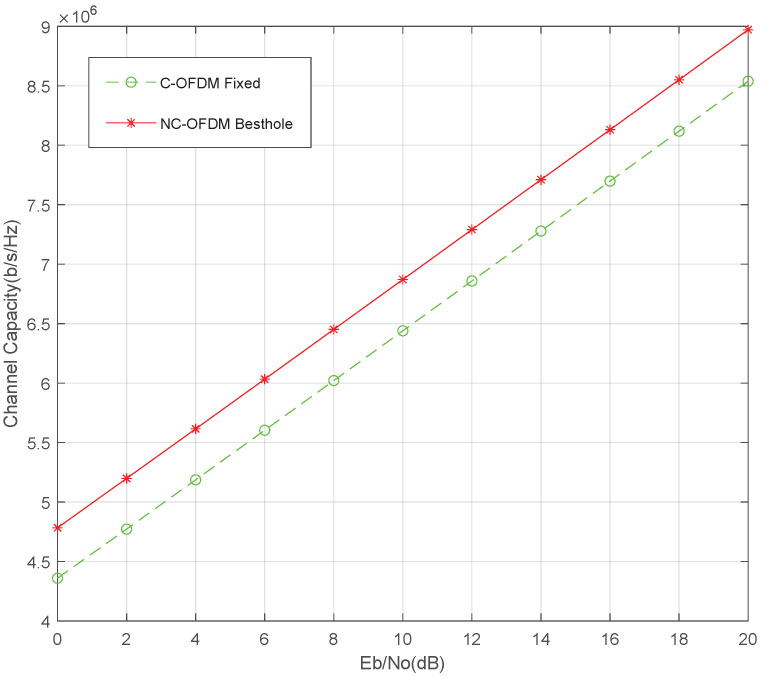
Comparison of channel capacity between NC-OFDM and C-OFDM.

**Figure 8 sensors-25-05596-f008:**
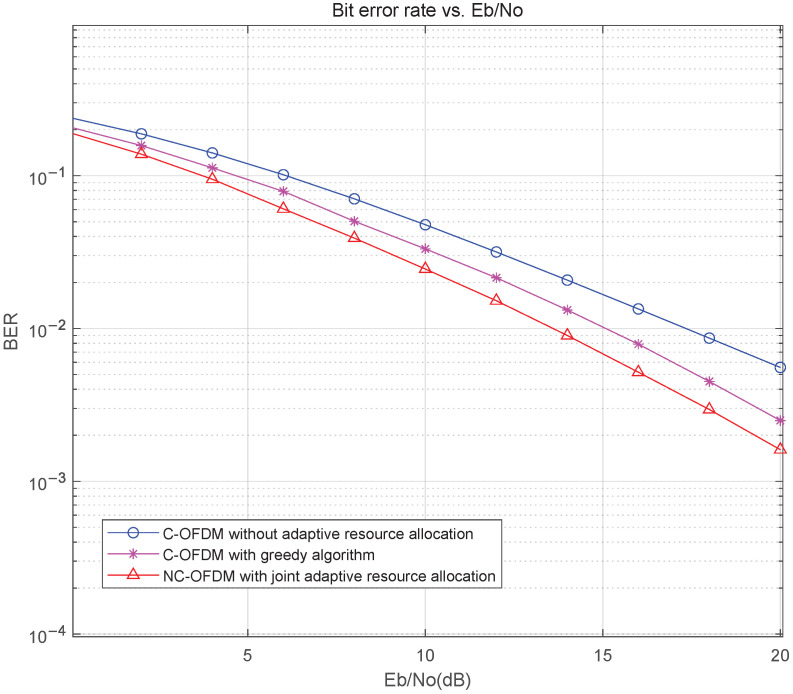
The BER performance comparison of NC-OFDM and C-OFDM.

**Figure 9 sensors-25-05596-f009:**
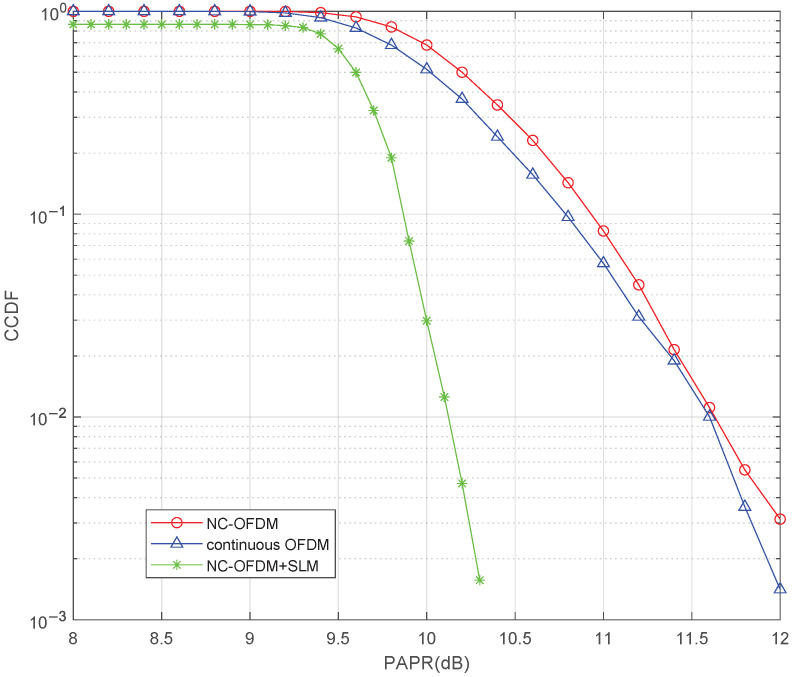
The PAPR performance comparison of NC-OFDM and C-OFDM.

**Figure 10 sensors-25-05596-f010:**
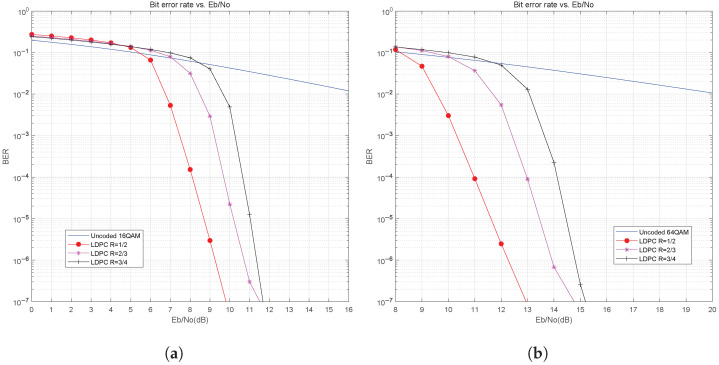
BER performance of LDPC with different QAM modulation: (**a**) 16QAM. (**b**) 64QAM.

**Figure 11 sensors-25-05596-f011:**
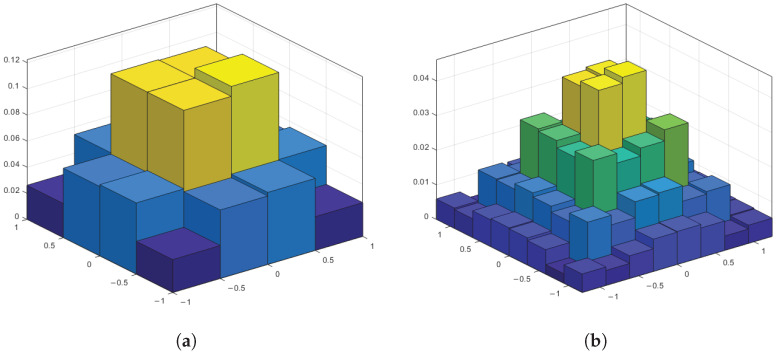
Probability distribution of constellation symbols: (**a**) 16QAM constellation. (**b**) 64QAM constellation.

**Figure 12 sensors-25-05596-f012:**
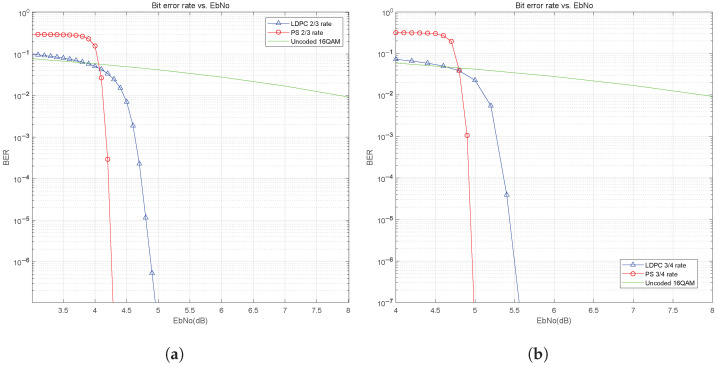
BER performance of PS for 16QAM in AWGN channel: (**a**) 2/3 code rate. (**b**) 3/4 code rate.

**Figure 13 sensors-25-05596-f013:**
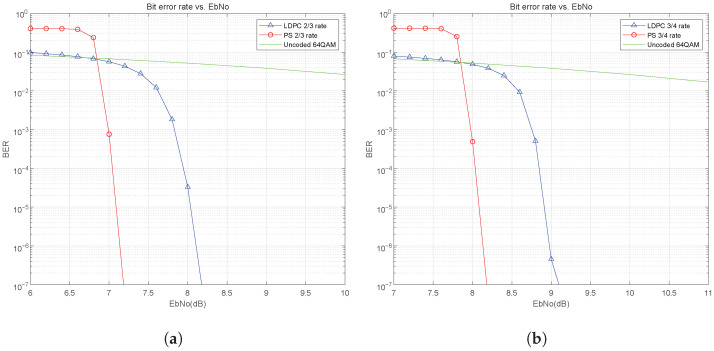
BER performance of PS for 64QAM in AWGN channel: (**a**) 2/3 code rate. (**b**) 3/4 code rate.

**Figure 14 sensors-25-05596-f014:**
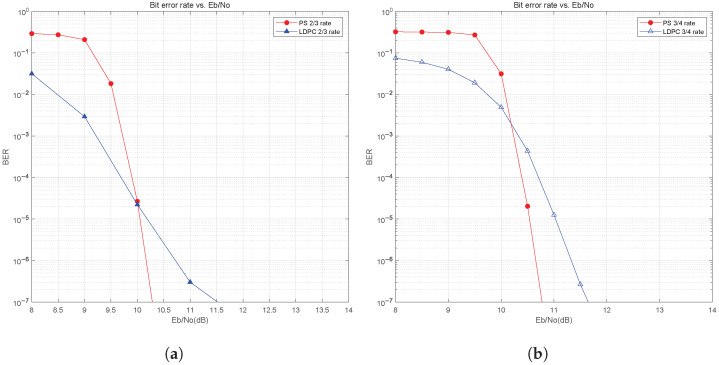
BER performance comparison of PS and LDPC for 16QAM in ITS channel: (**a**) 2/3 code rate. (**b**) 3/4 code rate.

**Figure 15 sensors-25-05596-f015:**
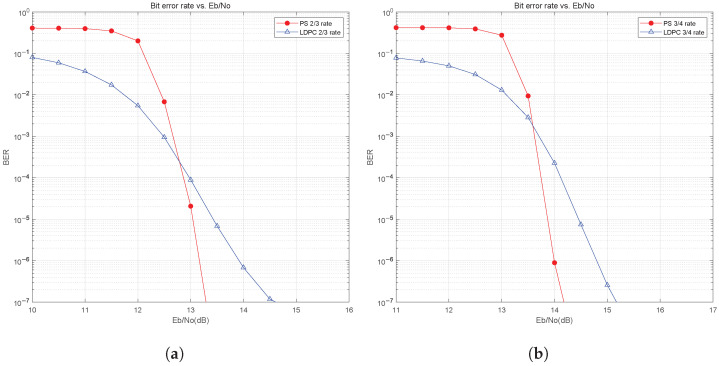
BER performance comparison of PS and LDPC for 64QAM in ITS channel: (**a**) 2/3 code rate. (**b**) 3/4 code rate.

**Table 1 sensors-25-05596-t001:** Comparison of the wideband HF communication solutions.

Solutions	Waveforms	Dynamic Spectrum Sensing	Spectral Efficiency
SC HF [10]	Single Carrier	No	Low
SC-CR HF [12]	Single Carrier	Yes	Medium
OFDM HF [7]	C-OFDM	No	Medium
Our proposed	NC-OFDM	Yes	High

**Table 2 sensors-25-05596-t002:** Main simulation parameters.

Parameter	Value
Frequency band	16 MHz∼23 MHz
Maximum bandwidth	1 MHz
Signal bandwidth	500 KHz
Channel maximum delay	70 us
Channel maximum doppler	0.1 Hz
Subcarrier spacing	39.0625 Hz
CP length	6.4 ms
FFT/IFFT size	25,600
QAM modulation	16QAM, 64QAM
System code rate	2/3, 3/4
LDPC code length	64,800
LDPC code rate	4/5
CCDM code rate	11/15, 9/10 ^1^ or 8/5, 37/20 ^2^

^1^ 16QAM. ^2^ 64QAM.

## Data Availability

All data are available to any researcher upon request.

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
