# Peer review of "Designand Implementation of a Novel Wideband HF Communication System Based on NC-OFDM and Probabilistic Shaping"

_sensors, 2025, doi:10.3390/s25175596_

Round 1

Reviewer 1 Report

Comments and Suggestions for Authors

This paper designs and verifies a broadband HF communication system combining NC-OFDM and probability shaping, which solves the problems of limited bandwidth and insufficient reliability of traditional HF communication. Its application value lies in improving the data rate and reliability of HF communication and supporting high-speed applications such as video transmission. However, some improvements are needed for the following issues:
1. The introduction part can compare the existing solutions in the relevant literature to highlight the advantages of NC-OFDM and PS in terms of spectrum efficiency and bit error rate.
2. The transmitter and receiver structure of NC-OFDM and the coding modulation scheme of probability shaping are described in the system model part. It is recommended to supplement the specific implementation algorithm and real-time analysis of dynamic spectrum sensing to ensure the system's adaptability in actual interference environments.
3. The experimental results can illustrate the effectiveness of NC-OFDM and PS. However, in order to further illustrate the robustness of the system, time-varying interference scenarios can be simulated, such as burst interference leading to subcarrier switching.

Author Response

Thank you very much for taking the time to review this manuscript. Please find the detailed responses in the attachment word file.

Reviewer 2 Report

Comments and Suggestions for Authors

This article presents a new approach on wideband HF communication system capable of supporting high-data-rate transmissions such as video over congested HF bands using NC-OFDM and providing more reliable and efficiency through probabilistic shaping combined with LDPC coding.

One of the original ideas behind this research is introducing NC-OFDM with probabilistic shaping-based LDPC coding for wideband HF communication — a combination that not been utilized previously in the current technologies. It addresses supporting high-throughput applications over unreliable and fragmented HF channels, where continuous bandwidth allocation is not feasible.

Compared to prior research that either focused on narrowband OFDM in HF or applied NC-OFDM, this paper extends the application to long-distance HF channels using a realistic ITS channel model. It also introduces a system design proved by simulation, showing improved BER performance and shaping gains, especially with high-order modulation schemes.

While the simulation-based approach provided in the paper is solid, the work could benefit from experimental validation under real HF channel conditions. Additionally, a more detailed discussion on implementation complexity particularly regarding FFT scaling and the overhead of probabilistic shaping may improve the work. 

The conclusions in the paper are consistent with the simulation data, clearly showing the benefits of NC-OFDM and PS in BER performance, shaping gain, and effective spectrum utilization. All key questions are addressed through proper simulations under ITS HF channel models, including comparative BER analyses of uncoded, LDPC-coded, and PS-LDPC coded systems across various modulation orders.

The references are relevant and include foundational work in HF communication, OFDM, LDPC, and probabilistic shaping. However, the paper would benefit from citing more recent experimental or real-world implementations of similar systems to better position its contribution.

The figures are good and clearly illustrate the system architecture, simulation results, and spectral characteristics. Nevertheless, some figures would benefit from additional annotations to clarify parameters and highlight key observations (e.g., exact shaping gain points, interference band locations). 

Overall the article can be accepted in present form.

Author Response

(The authors gave the same response as above.)

Reviewer 3 Report

Comments and Suggestions for Authors

This paper presents a novel transmission concept that has been implemented in modern systems such as 5G, Wifi 6 and that may also be applicable to wideband RF communications. The development of new broadband systems in the HF range is topical and the objective of this paper falls within the areas of interest in radio communications research. However, for this paper to be published, I propose the following recommendations:
1. Revise the English writing and expression. (many linguistic, expression and punctuation errors).

2. Imbalance between NC-OFDM representation and LDPC coding. A lot of theoretical data about LDPC coding and little about NC-OFDM (how this technique is used in other types of transmissions, what subcarrier assignment algorithms are used, advantages and disadvantages, etc.).

3. The materials and methods used in the simulation tests are not clearly and unambiguously presented (which simulation software was used, screenshots showing the configuration of the parameters presented, how the measurements were performed). Statements such as "for example, when 16QAM modulation is used, the parameters are" and nothing is specified for 64QAM can be confusing if the parameters were the same or changed.

4. In the Results and Discussion section, a very brief PAPR analysis is presented without the theoretical part of the paper. Comparative analysis between different code rates with or without PS and without coding is also presented, but the OFDM/NC-OFDM transmission technique is not discussed in these analyses.

5. The own contributions of this work to the development of such a system are not emphasized. It should be introduced what novelties this work brings to the field and what improvements/new features it introduces compared to other specialized works in the same field (some of them are listed in the references).

Author Response

(The authors gave the same response as above.)

Reviewer 4 Report

Comments and Suggestions for Authors

The authors proposed a wideband HF communication system based on non-contiguous OFDM  and probabilistic shaping, which can be used to enhance transmission reliability for video applications. I have a few comments below:

(1) There are some careless grammar issues in the manuscript. For example, line 18, " which can not only support ..."; Line 35, ". In the Ref. [4 ] studied the design..."

(2) Is there any difference of the NC-OFDM solutions  between satellite communications and this work except for the oprating requency?

(3) This work focuses on intra-system comparisons but lacks benchmarks against state-of-the-art HF communication schemes.

Author Response

(The authors gave the same response as above.)

Round 2

Reviewer 3 Report

Comments and Suggestions for Authors

This version of the manuscript has major improvements and is suitable for publication without changes.

Author Response

Thank you very much for appreciating the value of our work.

Reviewer 4 Report

Comments and Suggestions for Authors

The authors didnot answer my question:" Is there any difference of the NC-OFDM solutions between satellite communications and this work except for the oprating requency?" I do know that the bandwidth and the propagation environment are different between  HF communication and satellite communication. I mean the difference or innovations of this work. Since NC-OFDM solutions have been applied in satellite communications.

Author Response

(The authors gave the same response as above.)
